

# A new non-resonant laser-induced fluorescence instrument for the airborne in situ measurement of formaldehyde

Jason M. St. Clair[1,2], Andrew K. Swanson[1,3], Steven A. Bailey[1], Glenn M. Wolfe[1,2], Josette E. Marrero[4], Laura T. Iraci[4], John G. Hagopian[5,6], Thomas F. Hanisco[1]

[1]Atmospheric Chemistry and Dynamics Laboratory, NASA Goddard Space Flight Center, Greenbelt, MD, 20771, USA
[2]Joint Center for Earth Systems Technology, University of Maryland Baltimore County, Baltimore, MD, 21228, USA
[3]Goddard Earth Sciences Technology and Research, Universities Space Research Association, Columbia, MD, 21046, USA
[4]Atmospheric Science Branch, NASA Ames Research Center, Moffett Field, CA, 94035, USA
[5]Lambda Consulting/Advanced Nanophotonics, Harwood, MD, 20776, USA
[6]Optics Branch, Instrument Systems and Technology Division, NASA Goddard Space Flight Center, Greenbelt, MD, 20771, USA

*Correspondence to*: Jason M. St. Clair (jason.m.stclair@nasa.gov)

**Abstract.** A new in situ instrument for gas-phase formaldehyde (HCHO), COmpact Formaldehyde FluorescencE

Experiment (COFFEE), is presented. COFFEE utilizes non-resonant laser-induced fluorescence (NR-LIF) to measure HCHO, with 300 mW of 40 kHz 355 nm laser output exciting multiple HCHO absorption features. The resulting HCHO fluorescence is collected at 5 ns resolution, and the fluorescence time profile is fit to yield the ambient HCHO mixing ratio. Typical 1-σ precision at ~0 pptv HCHO is 150 pptv for 1 s data. The compact instrument was designed to operate with minimal in-flight operator interaction and infrequent maintenance (1-2 times per year). COFFEE fits in the wing pod of the

Alpha Jet stationed at NASA Ames Research Center and has successfully collected HCHO data on 27 flights through 2017 March. The frequent flights, combined with a potentially long-term data set, makes the Alpha Jet a promising platform for validation of satellite-based column HCHO.

## 1 Introduction

Formaldehyde (HCHO) is an abundant, photochemically influential trace species in the Earth's atmosphere. Primary sources of HCHO include biomass burning (Akagi et al., 2011; Andreae and Merlet, 2001) and fossil fuel combustion (Anderson et al., 1996; Luecken et al., 2012; Olaguer et al., 2009), but these are dwarfed by secondary production from the photochemical oxidation of volatile organic compounds (VOC). This secondary source is dominated by the locally abundant VOC(s): $CH_4$ in the remote atmosphere, isoprene in biogenically active regions (Palmer et al., 2003; Shim et al., 2005), and

unsaturated (Parrish et al., 2012) VOCs in regions with large anthropogenic VOC emissions. HCHO loss occurs via photolysis and reaction with OH, resulting in a daytime atmospheric lifetime of a few hours. Mixing ratios of HCHO vary



from tens of parts per trillion (pptv) in the remote atmosphere (Fried et al., 2003) to a few parts per million by volume (ppmv) in biomass burning plumes (Akagi et al., 2014), with typical values in the 50 pptv to 10 ppb range. Elevated HCHO, due to its limited atmospheric lifetime, is indicative of recent VOC oxidation, and in the upper troposphere and lower stratosphere (UT/LS) it suggests recent convective transport (Apel et al., 2012; Fried et al., 2008b, 2016). Measurements of

HCHO are valuable both as a tracer of recent VOC oxidation, and also due to its role in $HO_x/O_3$ chemistry (Jaeglé et al., 2001).

Atmospheric HCHO is measured using a variety of airborne instrumental methods, including mass spectrometry (Warneke et al., 2011), wet chemistry (Aiello and Mclaren, 2009; Junkermann and Burger, 2006; Lazrus et al., 1988), absorption

spectroscopy (Baidar et al., 2013; Catoire et al., 2012; Richter et al., 2015; Washenfelder et al., 2016; Weibring et al., 2006; Yokelson et al., 1999), and laser-induced fluorescence (LIF) (Cazorla et al., 2015; Hottle et al., 2009; Mohlmann, 1985). In addition to airborne observations, total column HCHO is measured by satellite (Chance et al., 2000; Steck et al., 2008), making HCHO one of the few VOCs observable from space. Numerous measurement technique reviews and instrument intercomparisons are available (Fried et al., 2008a; Hak et al., 2005; Kaiser et al., 2014; Zhu et al., 2016).

Traditionally, LIF measurements of HCHO have used a wavelength-tunable excitation laser to dither on and off the HCHO absorption feature, using the difference in signal to calculate the HCHO mixing ratio. The benefit of this approach is that the differential signal excludes any broadband background fluorescence from interfering with the HCHO measurement. The downside is that it requires either a large laser system unsuited for compact airborne instrumentation (Mohlmann, 1985), or a

custom, high cost fiber laser (Cazorla et al., 2015; Hottle et al., 2009). We present a new approach to measurement of HCHO by non-resonant laser-induced fluorescence (NR-LIF), using a fixed wavelength UV industrial laser at 355 nm to excite multiple HCHO absorption features simultaneously. Lacking the tunability and narrow linewidth necessary to dither on and off a single absorption feature, selectivity to HCHO is instead obtained using specialized fluorescence optical filters, and by employing high temporal resolution data acquisition to uniquely identify HCHO by its characteristic fluorescence

lifetime.

The new NR-LIF HCHO instrument, COmpact Formaldehyde FluorescencE Experiment (COFFEE), was designed specifically to join the payload of the Alpha Jet Atmospheric eXperiment (AJAX) out of NASA Ames Research Center in Mountain View, CA. The robust optomechanical design of the COFFEE instrument, combined with its simple and reliable

operation, makes the instrument ideal for long-term deployment to NASA Ames with minimal maintenance. The routine, long-term nature of the AJAX project, with flights approximately every two weeks, makes the Alpha Jet a good platform for monitoring seasonal and long-terms trends, as well as providing an extensive in situ data set for satellite validation.



## 2 Measurement technique

The COFFEE instrument uses NR-LIF for the detection of HCHO. Previous LIF-based instruments for atmospheric HCHO, such as the NASA In Situ Airborne Formaldehyde (ISAF) instrument (Cazorla et al., 2015), have used a narrow bandwidth, state-specific tunable excitation laser to target a specific absorption feature. COFFEE, in contrast, employs a moderate

bandwidth (full width at half maximum (FWHM) ~ 1 nm) fixed wavelength laser that excites multiple HCHO absorption features. The HCHO absorption cross section from Co et al. (2005), averaged to 0.001 nm resolution, is shown with the overlapping COFFEE laser output in the top panel of Fig. 1. The commercial off-the-shelf fixed wavelength laser is both less expensive and operationally more reliable than the narrow bandwidth tunable laser. In practice, the laser is turned on at the beginning of the flight and off at the end, with no other interaction.

HCHO fluorescence occurs over the ~355-550 nm wavelength range, as shown in Fig. 1, bottom panel. COFFEE has two detectors for collecting the banded fluorescence. Optical filter details are included in Sect. 3.2. Detection axis 1 uses a band pass filter centered at 450 nm (Fig. S1) to collect as much HCHO fluorescence as possible while excluding the primary sources of background counts (chamber, Raman, and Rayleigh scatter). Detection axis 2 (Fig. 1, bottom panel) utilizes a

multi-band pass filter that selectively transmits at HCHO fluorescence wavelengths, maximizing detection selectivity at the expense of decreased sensitivity.

Other LIF-based instruments for atmospheric HCHO (Cazorla et al., 2015; Hottle et al., 2009) collect fluorescence using a long pass filter to exclude scatter, and achieve measurement selectivity by alternately tuning the narrow bandwidth laser on

and off a HCHO absorption feature. The fixed wavelength laser in COFFEE cannot provide on and off line measurements. Measurement specificity to HCHO is instead achieved by acquiring the time-resolved fluorescence signal, 5 ns bins for 500 ns, and leveraging the unique fluorescence lifetime of HCHO in data processing. The details of data acquisition and data processing are discussed in Sect. 3.4 and 3.5, respectively.

## 3 Instrument description

### 3.1 Laser

A Spectra-Physics Explorer (EXPL-355-300-E, Fig. 2, item A) provides 300 mW of 355 nm of pulsed radiation at 40 kHz (adjustable 20-60 kHz). The laser is actively Q-switched, with a Nd:YVO$_4$ gain medium pumped by a single 808 nm diode to provide the 1064 nm fundamental wavelength. UV output at 355 nm is created using two intracavity lithium triborate

crystals for second and third harmonic generation. The pulse width (FWHM) is <15 ns, and the bandwidth is ~ 1 nm. The





laser is compact, with head dimensions 16.5 cm x 9.5 cm x 5.4 cm (0.9 kg) and power supply dimensions 16.4 cm x 13 cm x 6.6 cm (1.2 kg). Computer control is via RS232.

The laser head requires proper thermal management for the laser to perform to specification. 40 W of heat must be removed from the laser head at its maximum operating temperature of 308 K. Two thermal electric cooler (TEC) devices (TE Technology) provide thermal control of the laser head. The laser side of the TECs is controlled to 303 K, and the other side of the TECs are in thermal contact with the optical plate and heat sinks mounted to the underside of the optical plate.

## 3.2 Optical system

The optical layout of the instrument (Fig. 2, item B) is shown in more detail in Fig. 3. The entire optical system is contained on the optical plate in a single plane. The plate was machined out of 13 mm thick 6061 aluminum and is secured to the chassis at four points utilizing Sorbothane vibration isolation bushings. The plate is heated to 303 K.

    The laser beam is directed by two antireflection (AR)-coated dielectric mirrors (CVI Laser Optics) into the detection cell. A
collimating lens (F= 100 mm, Thorlabs) and a λ/2 wave plate (AR-coated, OptiSource), the latter used to minimize the Raman scattering directed at the photomultiplier tubes (PMTs), are positioned in between the turning mirrors. The detection cell is very similar to the cell in ISAF (Cazorla et al., 2015), with the main differences being the number and orientation of the PMTs, and the optical filters used. The beam enters and exits the cell through AR-coated fused silica windows (CVI Laser Optics) that are mounted at a 3.5° angle to prevent surface reflections from reaching the PMTs. Inside the cell, the
beam continues through a series of circular baffles, 4 before the detection volume and 3 after, which drastically reduce stray light. The baffle apertures are progressively larger along the beam propagation path (2.5 mm, 3.0 mm, 3.5 mm). The baffles adjacent to the detection volume are coated with a carbon nanotube coating (Hagopian, 2011); the other baffles are laser-cut and painted black (Lenox Laser). The interior of the detection cell is coated with a molybdenum oxide treatment (Insta-Black 380, EPI), to further eliminate stray light.


    On two sides of the detection cell, aspheric lenses (NA=0.66, AR-coated, Edmund Optics) image the volume where the laser beam and main gas flow cross. From each lens, the image is reflected 90° by a turning mirror (right angle prism dielectric, Thorlabs) and passed through a series of optical filters before being partially focused by a lens (F = 75 mm, AR-coated, Thorlabs) onto a photomultiplier tube (PMT) (Hamamatsu H7360-02 MOD). The optical filters differ between the two
PMTs. Arranged in order from the aspheric lens to the PMT, axis 1 contains an AR-coated 370 nm long pass absorption filter (Hoya Candeo Optronics), a 450 nm, 70 nm wide band pass interference filter (Semrock), and a 395 nm long pass absorption filter (Edmund Optics). Axis 2 uses a 400 nm long pass interference filter (Omega Optical), an AR-coated 370





nm long pass absorption filter (Hoya Candeo Optronics), a custom 11-band band pass interference filter (Semrock), and a 395 nm long pass absorption filter (Edmund Optics). The 11-band filter was designed to selectively transmit formaldehyde fluorescence while reducing the background. After the detection cell, a beam sampler (Thorlabs) splits the beam, and the main beam continues to a beam dump. The beam sampler reflection is directed to a power meter consisting of a diffuser

(Thor DGUV10-600), absorption filter (Thorlabs FGUV11), and an amplified photodiode (OSI 555-UV).

### 3.3 Gas handling

The fundamental design consideration for the instrument sample flow is to minimize the potential for the adsorption/release of HCHO to/from exposed surfaces (Cazorla et al., 2015; Wert et al., 2002). To that end, all surfaces that deliver gas to the

detection cell are either fluorocarbon (FEP, THV) or fluorocarbon-coated (FluoroPel, Cytonix). The current Alpha Jet inlet is a rear-facing stainless steel tube 9.5 mm OD (6 mm ID) that extends 17 cm beyond the bottom of the pod. 9.5 mm OD (6.35 mm ID) THV fluoropolymer tubing connects the inlet to the instrument chassis. The instrument is operated with an inline particle filter (Balston 9922-05-DQ) when high aerosol loading is expected in order to minimize related measurement artifacts (see Sect. 4.5). The filter housing is Kynar (polyvinylidene fluoride) and the filter element is a microfiber with a

fluorocarbon resin binder. The element retains 93% of the particles with a 0.01 μm diameter.

Inside the instrument, 5 cm of 9.5 mm OD PFA tubing connects from the chassis to a pressure controller, and 15 cm of 9.5 mm OD PFA tubing connects from the pressure controller to the detection cell. The pressure controller (Fig. 2, item C) is an actuator (iQ Valve) coupled with a custom valve block, and is heated to 308 K. The detection cell pressure is regulated to

10.7 kPa. The main flow passes directly down through the detection cell and out of the chassis to the vacuum pump (Vacuubrand MD-1, Fig. 2 item D). A small amount of air is pulled through the laser baffle arms to flush that volume, and the flow is combined with the main flow (after the detection cell) before exiting the chassis. In lab, the instrument sampling flow is 2.3 sLm.

### 3.4 Data acquisition

Data acquisition and instrument control is conducted by a National Instruments CompactRIO system, hereafter RIO (Fig. 2, item E). The RIO consists of a main processor module (running a realtime operating system) and a backplane driven by a field programmable gate array (FPGA). Additional plugin modules add I/O. NI 9205 and NI 9264 modules provide analog input and output, respectively. Two channels of a NI 9402 high speed digital I/O module are programmed as 5 ns resolution

counters, with each PMT having its own counter. The counters are triggered by the OptoSync from the laser (30-100 ns after the laser pulse), which provides a TTL pulse closer in coincidence with the laser pulse than obtained from the laser trigger





out. In order for the PMT signals to arrive after the counters are triggered, they are delayed by 50 ns with a passive delay circuit (Data Delay Devices, 1515 series).

Data for each PMT channel are acquired in two ways: 1) integrated every 0.1 s with non-gated (continuous) and gated data
streams, which are used primarily for diagnostic purposes, and 2) integrated every 1 s and time-resolved to 100 discrete time bins, each 5 ns wide, that cover the 500 ns immediately following the counter trigger. The 5 ns time-resolved data are the key to the data processing approach necessary to minimize measurement artifacts with the NR-LIF approach, as will be discussed in Sect. 3.5, and is used to produce the HCHO mixing ratio data product. Diagnostic data (laser power, pressures, temperatures, etc.) are also recorded every 1 s.

**3.5 Data processing**

HCHO mixing ratios are obtained using the 5 ns bin time-resolved profiles from the two detection axes. The data processing consists of three steps, each done independently for the two detection axes: 1) subtraction of the minor 'long-lived' component from the time profile; 2) two-parameter nonlinear least squares fit of the data using profiles (hereafter referred to as exemplars) that represent the HCHO and non-HCHO (chamber scatter, Raman and Rayleigh scatter, fluorescence of
optics, etc.) contributions to the observed profile; 3) one-parameter nonlinear least squares fit with the non-HCHO contribution fixed from the previous two-parameter fit, and only the HCHO contribution allowed to vary. The second pass fit with only one parameter improves the precision of the measurement.

**3.5.1 Long-lived component**

The fluorescence signal at the end of the bin-resolved data (~400 ns after the laser pulse) is small but non-zero, and changes in this 'long-lived' signal do not scale with changes in the non-HCHO 'air' exemplar, necessitating a separate treatment. The long-lived signal has a longer fluorescence lifetime than HCHO, which permits fitting and removal of the long-lived signal without interference from ambient HCHO. For detection axis 1, an empirical profile determined from a laboratory run is scaled to fit the observed 1 Hz data using a single parameter least squares fit to the observed profile from bin 75 to bin
100, and the scaled profile is subtracted from the observed data before performing the exemplar fits. For detection axis 2, the long-lived signal is smaller than for axis 1 by a factor of ~ 6 and is stable over the last ~15 bins, and so a simpler treatment is used: the observed 1 Hz profile is averaged from bin 87 to bin 100 and the mean is subtracted as a constant from all bins in the observed profile before performing the exemplar fits. All fitting and exemplar creation is done using data with the long-lived component removed.



### 3.5.2 Exemplar fits

**Obtaining the exemplars**

The representative time profiles, or exemplars, are determined from laboratory calibration runs where the instrument samples clean, dry air (typically UHP dry air) with varied amounts of HCHO added.  The 'air exemplar', which represents all non-HCHO contributions to the observed profile, is obtained by time-averaging the observed profile when no HCHO is added to the dry air.  Figure 4 shows the profiles involved in creating the 'HCHO exemplar'.  The HCHO exemplar (Fig. 4, red dashed) is obtained by time averaging the observed profile (Fig. 4, cyan circles) during the calibration period of maximum
HCHO (typically 25-30 ppbv) and subtracting the air exemplar (Fig. 4, blue dashed) from the time-averaged profile.  The highest HCHO period is used so that HCHO dominates the shape of the observed profile.

**Two-parameter exemplar fit**

An example two-parameter exemplar fit is shown in Fig. 5.  The observed profile (Fig. 5, cyan circles), with the long-lived
component removed, is fit with a linear combination of the air exemplar and the HCHO exemplar.  The fit parameters are the scalar multipliers applied to the exemplars: the scaled air exemplar (Fig. 5, blue dashed) and scaled HCHO exemplar (Fig. 5, red dashed).  The least squares optimization is performed on the data from bin 13 to bin 60, with the fit window chosen to maximize data precision and fit quality, as determined by visual inspection of fit residuals.  The optimized fit for the bin 13-60 window is shown in black.

**One-parameter exemplar fit**

The first step of the one-parameter fit applies a 21 s median filter to the vector of air exemplar fit scalars from the two-parameter fit.  The smoothed vector is then used in a one-parameter fit where the air exemplar contribution is fixed to the air exemplar scaled by the smoothed vector, and the HCHO exemplar scaling factor is allowed to vary.  The result is a higher
precision fit, and is possible because the phenomena that comprise the 'air exemplar' contribution to the observed profile (chamber scatter, Raman and Rayleigh scatter, fluorescence of optics, etc.) do not change rapidly.  The output of the one-parameter fit, the HCHO exemplar scalar, is directly proportional to HCHO mixing ratio.  HCHO data in pptv are obtained by applying a calibration factor, unique to the HCHO exemplar used, to the fit output.  The final HCHO mixing ratio data product is produced by averaging the data from the two detection axes.




### 3.5.3 Data processing with gated count data

In addition to data processing with exemplar profiles, time-gated 1 Hz data derived from the time-resolved profiles can be used to obtain HCHO, with higher measurement precision than is achieved with the exemplar fits. For example, a calibration experiment yields, for 1 s data and 0 pptv added HCHO, standard deviations of 150 pptv for the one-parameter

exemplar fit (175 pptv for the two-parameter fit only) and 130 pptv for the gated count data (167 pptv for ungated count data). The time-gated data excludes much of the prompt signal from scatter by summing counts from bin 24 to bin 100 (115 ns to 500 ns). Using the same laboratory calibration experiment as an example: with no HCHO added, the gate excluded 89% (450 nm filter detection axis) and 95% (multi-band pass axis) of the total signal in the first 500 ns from the trigger. More of the HCHO signal is retained due to its fluorescence lifetime: 73% of the HCHO signal is excluded by the gate.

Gated count 10 Hz data, as well as ungated count 10 Hz data, can be used to obtain HCHO mixing ratios. The 10 Hz data are used only for diagnostic purposes, e.g. the instrument flush time experiment in Sect. 4.4.

The count signal is converted to HCHO mixing ratio using a linear relationship determined from laboratory calibrations, with the slope being the instrument sensitivity to HCHO (discussed in Sect. 4.1) and the intercept being the signal at HCHO = 0

pptv, comprised of the same signal sources as the air exemplar: chamber scatter, Raman and Rayleigh scatter, and fluorescence of optics. While the count-derived HCHO data are higher precision than the exemplar fit-derived HCHO data, the count-derived data are potentially more prone to measurement error from changes in background signal due to changes in alignment, degradation of optics, the presence of aerosol (Mie scattering), or from unknown fluorescing compounds. In contrast, ISAF (Cazorla et al., 2015) is immune to these changes in background due it its measurement of online and offline

signal. Currently the count-derived HCHO data are only used for diagnostic purposes.

## 4 Performance

### 4.1 Sensitivity

The sensitivity of each detection axis to a given amount of HCHO is a function of a number of instrument parameters: laser

power, collection optics efficiency, fluorescence optical filter transmission, and PMT response. As for ISAF, none of the instrument parameters that affect instrument sensitivity are expected to degrade on a time scale shorter than years. The HCHO calibration of the instrument has been measured 2-3 times per year, and will be measured at least once a year in the future to track any changes in sensitivity.

Calibration is performed using measured flows from two cylinders, one containing ultra-high purity (UHP) air further purified with a Drierite/molecular sieve scrubber and the other a ~500 ppbv mixture of HCHO in $N_2$. The exact



concentration of the HCHO mixture in all of our HCHO standard cylinders is measured yearly using IR absorption, with less frequent verification of the IR measurement by long-path UV absorption. Details of the HCHO cylinder assessment via IR and UV absorption are available in Cazorla et al. (2015). HCHO calibration accuracy for COFFEE is determined by the uncertainty in the HCHO standard concentration as well as the uncertainty in the gas flow dilution described below, and is

estimated to be ±10%.

For calibration, flow of the HCHO standard is sequentially set to 3-5 different flows in the range 0-50 standard cm$^3$ min$^{-1}$ (sccm) and is added to a carrier flow of UHP air, typically 3-5 standard L min$^{-1}$ (sLm). The instrument draws ~ 2.3 sLm and the remaining gas flow exhausts to the room before the pressure controller—the additional flow improves the time response

of the calibration system. Typical calibration data for detection axis 2 is shown in Fig. S2.

Instrument sensitivity to HCHO differs for the two detection axes primarily due to their respective optical filter transmission, with axis 1 more sensitive than axis 2. The gated count sensitivities for axis 1 and axis 2 are 0.29 counts s$^{-1}$ mW$^{-1}$ ppbv$^{-1}$ and 0.13 counts s$^{-1}$ mW$^{-1}$ ppbv$^{-1}$, respectively. The ungated sensitivities for axis 1 and axis 2 are 0.98 counts s$^{-1}$ mW$^{-1}$ ppbv$^{-1}$ and

0.47 counts s$^{-1}$ mW$^{-1}$ ppbv$^{-1}$, respectively. For comparison, the ISAF sensitivity is 75 counts s$^{-1}$ mW$^{-1}$ ppbv$^{-1}$ for its typical 100 mbar operating pressure. Power normalized sensitivities are significantly lower (>100x) for COFFEE than the ISAF instruments primarily due to the less efficient overlap of the COFFEE laser output with the HCHO absorption lines.

**4.2 Precision**

Measurement precision is the dominant component of overall measurement uncertainty at low (< 700 pptv) mixing ratios. The standard deviation using data from two laboratory calibration experiments is shown in Fig. 6. At [HCHO] = 0 ppbv, the precision is ±130 pptv in 1 s and ±60 pptv in 10 s. Relative measurement precision improves with increasing HCHO, as shown in Fig. S3 using the same calibration data. The largest source of noise for COFFEE HCHO is Raman and Raleigh scattering of the excitation beam by air. Chamber scatter accounts for ~15% of signal (2 counts s$^{-1}$ mW$^{-1}$) for axis 1 and

~25% of signal (1 count s$^{-1}$ mW$^{-1}$) for axis 2 at 10.7 kPa and HCHO = 0 ppbv, with the remaining signal from Raman and Raleigh scatter.

Precision should improve as data are time-averaged. In practice, the benefit of additional time-averaging ceases when the data variability is no longer dominated by random noise. The Allan deviation plot shown in Fig. 7 demonstrates this point

for COFFEE HCHO data from a laboratory calibration with ~5 ppbv HCHO, processed with the two-parameter exemplar fit. The precision of the HCHO data improves with averaging until reaching a 250 s averaging time basis, implying that the signal-to-noise ratio for COFFEE measurements will improve from 1 s data to 10 s data, and again to 1-minute data. Fitting the decreasing linear (in log-log space) portion of the data yields a slope of -0.5, which is consistent with the data variability



being dominated by white noise on time scales shorter than 250 s. The full fit-based data processing includes an additional step beyond the two-parameter fit, and a similar Allan deviation analysis gives a slope of -0.4. The difference is likely due to the median filtering applied to the air exemplar scalar before conducting the one-parameter fit.

5 **4.3 Measurement uncertainty**

The overall measurement uncertainty for COFFEE HCHO is estimated to be ± (20% of [HCHO] + 100 pptv). As discussed in Sect. 4.1, the calibration uncertainty is ±10% of [HCHO]. The additional 10% uncertainty is added to conservatively account for unquantified sources of error such as unknown signal sources other than HCHO and any fit biases. Additional 10 and more extensive opportunities for instrument intercomparison in situ will likely reduce the need for this extra uncertainty. The 100 pptv term is to account for changes in the background signal that are not fully accounted for by the scaled air exemplar when performing the two-parameter fit.

**4.4 Time response**

15 Instrument time response directly affects the ability to resolve fine structure in atmospheric HCHO, and can affect measurement accuracy in regions of high HCHO contrast such as biomass burning plumes. Understanding the instrument time response is critical to properly interpreting the in situ data. Assuming a volume of 60 cm$^3$ and a volumetric flow of 29 L min$^{-1}$ (2.3 sLm, 303 K, 10.7 kPa), the e-fold flush time of COFFEE was estimated to be 125 ms. The actual time response of the instrument was measured by introducing narrow time pulses of HCHO into the instrument and fitting the signal decay, 20 as shown in Fig. 8. A low volume, rapidly switching valve (The Lee Company, IEP series) provided a 10 ms pulse of HCHO every 2 minutes into a flow of UHP air. The HCHO signal after the pulse was fit with an exponential decay, yielding an empirical e-fold flush time of 170 ms, 45 ms slower than the flush time estimated from volume and flow rate alone. Typically the data rate reported by COFFEE is 1 Hz, and therefore the 170 ms 1/e instrument response time will have a very limited effect on the observed HCHO data.

25

**4.5 Measurement interference from aerosol**

Mie scattering from the presence of aerosol increases the prompt signal (< 75 ns after trigger) detected by COFFEE. The additional prompt signal complicates the exemplar fitting routine by altering the profile shape of the non-HCHO component in the data time profile. To avoid the presence of Mie scattering, all ambient sampling with COFFEE is conducted through 30 an in-line particle filter. An example of the error experienced by COFFEE from unfiltered ambient sampling is shown in Fig. 9. COFFEE and ISAF were installed in an office trailer on the roof of a NASA Goddard laboratory building, and both



instruments sampled ambient air through the same ¼" OD PFA tubing mounted on the roof of the trailer. With COFFEE sampling through the particle filter, the instruments agree well sampling ambient air, room air, and Drierite-filtered air. The measurements do not agree as well, with a difference of >1 ppbv, for periods of ambient sampling without a particle filter on COFFEE, shaded gray.

## 5 Field deployment

### 5.1 Alpha Jet integration

COFFEE was designed specifically for integration onto the Alpha Jet (H211, LLC) stationed at NASA Ames—Moffett Field to participate in the Alpha Jet Atmospheric eXperiment (AJAX) (Hamill et al., 2016). The Alpha Jet carries four wing pods,

with the outboard pods containing fuel and the inboard pods available for instrumentation. Each instrument wing pod has a usable volume of ~0.1 m$^3$ and a maximum payload weight of 136 kg. COFFEE mounts in the mid-body of the left wing pod, as shown in Fig. 10. The instrument chassis (Fig. 2, item F) and pump (Fig. 2, item D) attach to a rack designed for use in the wing pods (Fig. 2, item G) and the rack then slides into the pod mid-body, making removal of the instrument straightforward for infrequent maintenance.

### 5.2 Flight data

The first flight of COFFEE on the Alpha Jet was on 2015 December 15. Since then, COFFEE has operated on 27 AJAX flights through March 2017, with data coverage predominately in the Bay Area and Central Valley of California. Figure 11 shows a map with overlaid flight tracks for all the AJAX flights with COFFEE, and Table 1 lists the dates and objectives for

each of the 27 flights. During this period COFFEE returned to GSFC just three times for maintenance, typically timed to coincide with aircraft maintenance.

Data from AJAX Flight 185 on 2016 April 19 are shown in Fig. 12 as an example of COFFEE HCHO performance. The flight included two spiral profiles, one over the San Joaquin Valley near Merced, CA (37.38˚ N, 120.6˚ W) and one directly

west, offshore over the Pacific Ocean (37.17˚ N, 123.2˚ W). The altitude profiles of HCHO for the two spirals are shown in the bottom panel of Fig. 12 for 10 s data. The onshore and offshore profiles are similar between 4 km and 9 km, and diverge considerably below 2 km as local photochemistry drives HCHO production over land. The profiles serve as a demonstration of the data set available for validation of HCHO satellite retrievals using routine AJAX flights over targeted profile locations.




## 6 Summary

The NR-LIF technique utilized in COFFEE has proven to be a viable, operationally robust approach to measuring gas phase in situ HCHO. While not achieving the sensitivity of a state-selective LIF instrument such as ISAF (Cazorla et al., 2015),

the NR-LIF technique provides adequate precision (1-$\sigma$ of 150 pptv for 1 s data at 0 pptv HCHO) for most scientific pursuits, with a lower cost, highly reliable laser. COFFEE data from over two dozen AJAX flights spread over 15 months have demonstrated the potential utility of the aircraft platform for validation of satellite-based total column HCHO.

## 7 Data availability

AJAX data are available upon request. Data contact: Laura Iraci (Laura.T.Iraci@nasa.gov).

**The Supplement related to this article is available online.**

*Acknowledgements.* Funding was provided by the Goddard Internal Research and Development (IRAD) program and NASA (NNH16ZDA001N-UACO). J. E. Marrero gratefully acknowledges funding from the NASA Postdoctoral Program.

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

**Table 1: AJAX flight information with COFFEE in payload, through March 2017.**



| Flight date | AJAX flight number | Flight objective |
|---|---|---|
| 12/15/2015 | F178 | Test flight; San Joaquin Valley |
| 01/08/2016 | F179 | Aliso Canyon methane leak |
| 01/12/2016 | F180 | Railroad Valley for $CH_4/CO_2$ satellite validation |
| 03/17/2016 | F182 | Boundary layer sampling around Sacramento |
| 03/23/2016 | F183 | Aliso Canyon after methane leak capped |
| 03/30/2016 | F184 | Level boundary layer legs in the San Joaquin Valley |
| 04/19/2016 | F185 | San Joaquin Valley (onshore) and offshore vertical profiles |
| 04/26/2016 | F186 | Vertical profiles at Visalia, Panoche, and Bodega Bay |
| 05/04/2016 | F187 | Two onshore and one offshore vertical profiles |
| 05/12/2016 | F188 | Offshore vertical profiles under OCO-2 satellite |
| 06/15/2016 | F191 | Vertical profiles offshore (Bodega Bay) and onshore (Visalia); fire sampling |
| 06/21/2016 | F192 | Vertical profiles at Point Sur, Chews Ridge, Panoche, Visalia |
| 07/01/2016 | F193 | Railroad Valley for $CH_4/CO_2$ satellite validation |
| 07/21/2016 | F195 | Vertical profiles offshore near Bodega Bay and over land near Visalia |
| 07/28/2016 | F196 | Soberanes Fire |
| 08/09/2016 | F197 | Soberanes Fire |
| 08/12/2016 | F198 | Soberanes Fire |
| 08/24/2016 | F199 | Soberanes and Cedar Fires |
| 09/14/2016 | F200 | Soberanes Fire |
| 09/21/2016 | F201 | Vertical profiles and level legs in the San Francisco Bay Area: offshore, San Martin, Patterson Pass, Bethel Island |
| 11/02/2016 | F202 | Vertical profiles and level legs in the San Francisco Bay Area: Bodega Bay, Pittsburg/Delta, East San Jose |
| 11/03/2016 | F203 | Vertical profiles and level legs in the San Francisco Bay Area: offshore, Bethel Island, Livermore, East San Jose |
| 12/02/2016 | F204 | Chews Ridge and MMS calibration maneuvers |
| 02/23/2017 | F207 | Vertical profiles and level legs in the San Francisco Bay Area: Bodega Bay, Bethel Island, San Martin |
| 03/01/2017 | F208 | Vertical profiles and level legs in the San Francisco Bay Area: Bodega Bay, Bethel Island, San Martin, Livermore |
| 03/09/2017 | F209 | Boundary layer legs along Walker and Tehachapi Passes, Antelope Valley |
| 03/23/2017 | F210 | Boundary layer legs along San Pablo Bay, Sacramento River Delta, McDonald Island |



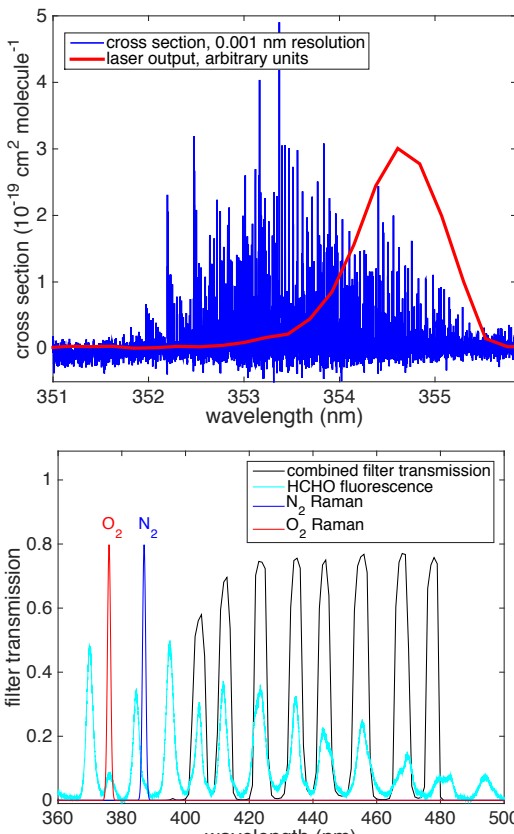

**Figure 1: Top Panel: The HCHO absorption spectrum (Co et al., 2005), averaged to 0.001 nm resolution, and the excitation laser spectrum are shown. Bottom Panel: The optical filter transmission spectrum is shown for the detection axis 2 (multi-band pass filter). The HCHO fluorescence and the N₂ and O₂ Raman spectra are included for reference, all with arbitrary units.**

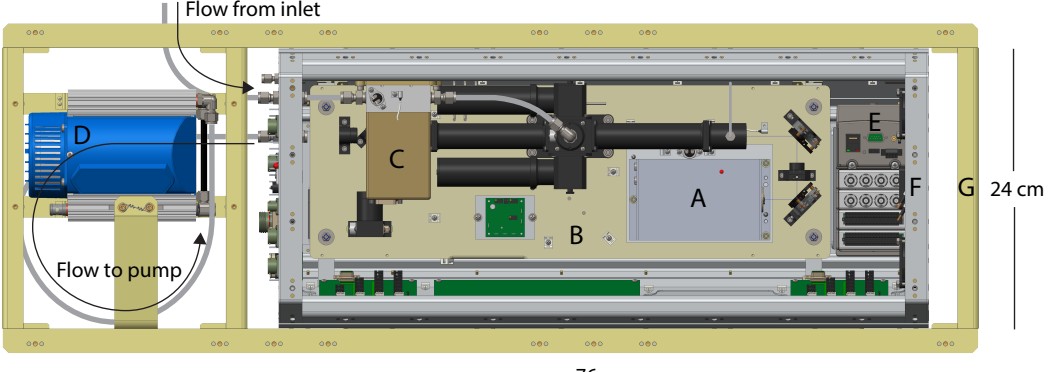

**Figure 2: COFFEE instrument layout in the AJAX pod rack, including (A) laser, (B) optical plate, (C) pressure controller, (D) vacuum pump, (E) RIO data acquisition system, (F) instrument chassis, and (G) AJAX pod rack.**

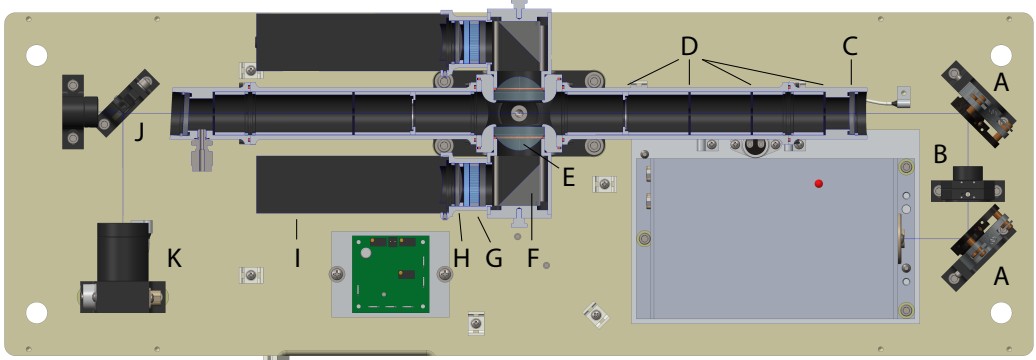

**Figure 3: The optical plate layout is shown, with a cut-away of the detection cell. The components include (A) steering mirrors, (B) half-wave plate and collimating lens, (C) cell windows, (D) laser baffles, (E) aspheric lens, (F) prism dielectric mirror, (G) optical filters, (H) lens, (I) photomultiplier tube, (J) beam splitter, and (K) laser power monitor.**





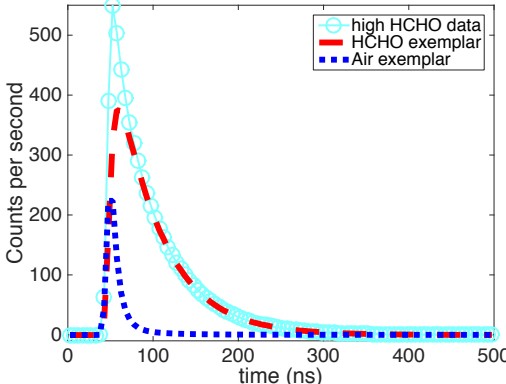

**Figure 4: Exemplar time profiles are obtained in the laboratory. The air exemplar is created by averaging the profile with no added HCHO (blue dashed line), and the HCHO exemplar (red dashed line) is obtained by subtracting the air exemplar from data**
5 **with high HCHO (cyan circles).**

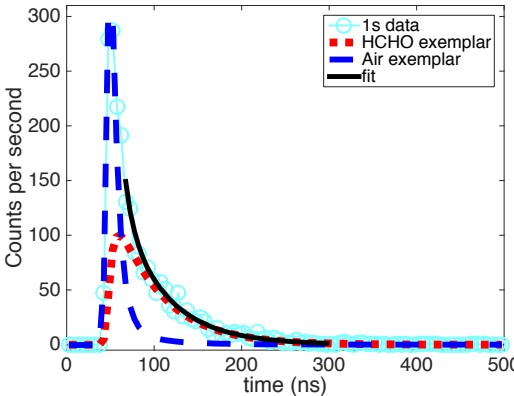

**Figure 5: Each 1s data profile (cyan circles) is fit using a linear combination of the air (dashed blue line) and HCHO (dashed red**
10 **line). The fit profile, over the time window used for the least squares fit, is shown in black.**





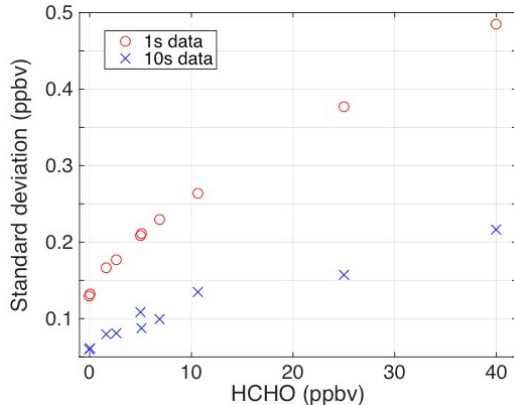

**Figure 6: The standard deviation as a function of HCHO is shown to demonstrate the precision of the HCHO measurement for 1 s and 10 s averaging.**

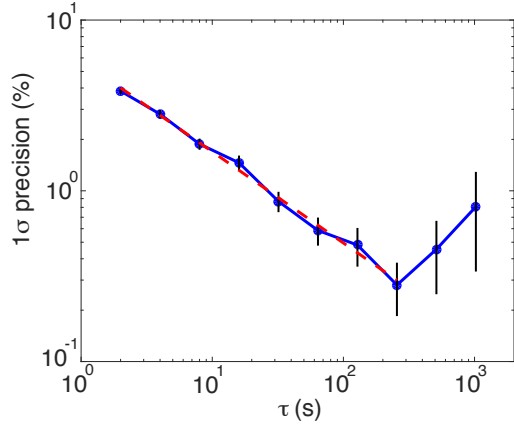

**Figure 7: Normalized Allan deviation as a function of averaging time (τ) demonstrates the precision benefit of time-averaging up to 250 s. Fitting the data with τ < 250 s yields a slope of -0.5 (dashed red line), consistent with white noise dominating the**

10    **variability at shorter averaging periods. [HCHO] = 5 ppbv.**



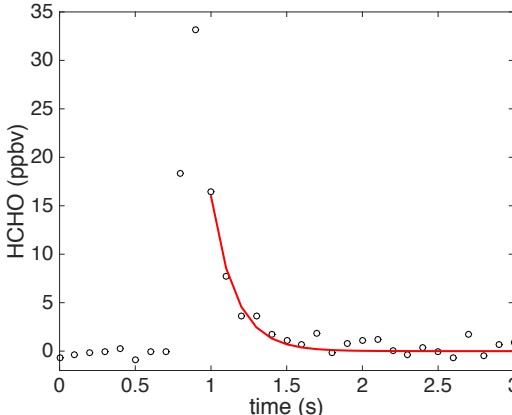

**Figure 8: Instrument time response to a pulse of HCHO is fit with an exponential decay, giving an empirical e-fold flush time of 170 ms.**

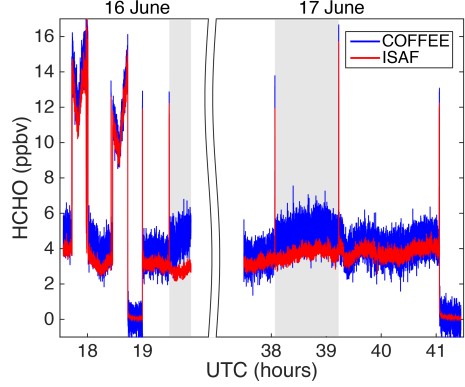

**Figure 9: COFFEE (blue) and ISAF (red) data sampling from the roof of Building 33 at GSFC in June 2015. Shaded sections indicate COFFEE sampling without a particle filter. HCHO above 8 ppbv was from sampling indoor air, ~ 4 ppbv was ambient sampling, and ~0 ppbv was through a Drierite/molecular sieve scrubber.**



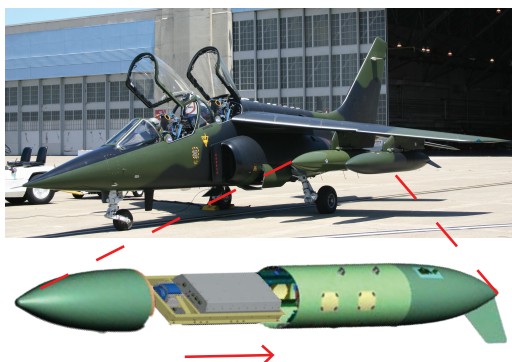

**Figure 10: The COFFEE instrument, installed in its AJAX pod rack, is mounted into the mid-body of the inboard left pod.**

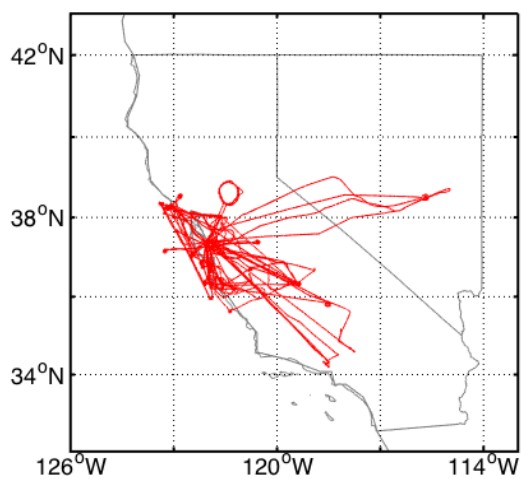

**Figure 11: Map of AJAX flight tracks with COFFEE in payload through March 2017.**



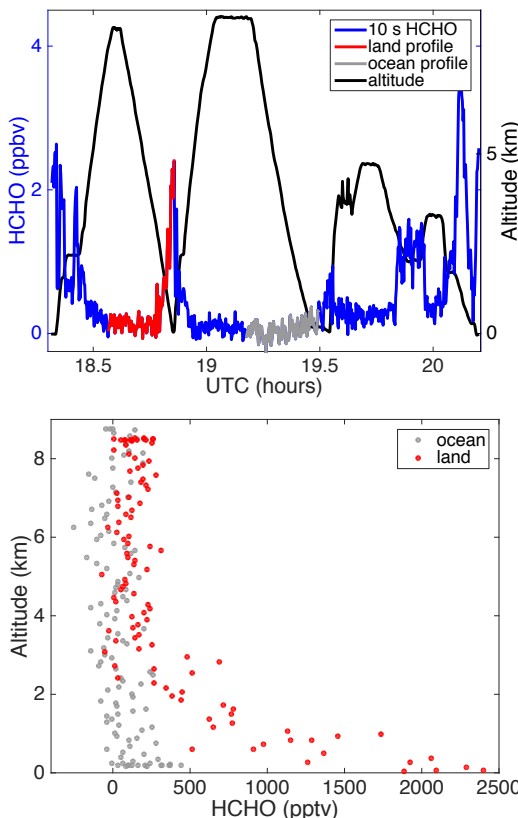

**Figure 12: Flight data for AJAX Flight 185 on 2016 April 19. Top Panel: time series of HCHO (10 s data, blue) and the corresponding aircraft altitude (black), with spiral profiles over land (red) and ocean (gray) highlighted. Bottom Panel: altitude profiles of HCHO (10 s data) over land (red) and ocean (gray).**