# Peer review of "A new non-resonant laser-induced fluorescence instrument for the airborne in situ measurement of formaldehyde"

_Atmospheric Measurement Techniques, 2017_

## Referee Comment (RC1) · Anonymous Referee #1 · 9 Sep 2017

General comments:

The manuscript entitled "A new non-resonant laser-induced fluorescence instrument for the airborne in situ measurement of formaldehyde", by J. M. St. Clair, et al., demonstrates a new instrument for measuring formaldehyde in aircraft campaigns. The paper describes in detail the technical aspects of the new LIF instrument, including the custom-built optical system, sampling technique, and the methods for processing the fluorescence time profiles. It is a very well written paper, and it carefully describes a novel instrument that should be of use to the atmospheric chemistry community. Publication is recommended, following the consideration of a few minor comments:

[Figure]

Specific comments:

Section 3.3 – Can you address what, if any, losses of HCHO are expected on the in-line particle filter (or even on the pressure controller)? Has this been tested? Were any laboratory calibrations conducted with the particle filters in place to make sure they match calibrations without filters?

P5L13 – Particle filters were used when "high aerosol loading is expected". Does this mean they were used in any non-laboratory deployment?

P5L15 – "The element retains 93% of the particles with a 0.01 um diameter." Does this mean it retains 93% of all particles larger than 0.01 um? Or is there some diameter over which it retains close to 100% of particles? What percentage of particle smaller than 0.01 um are retained, and do you expect this to make a difference in your analysis?

Section 3.3 – Is the inlet system here similar to the one shown in Fig. 5 of Carzorla et al? If not, describe the inlet used here. Is there anything in addition to the particle filter?

Section 3.5 –Clarify in the beginning of this section that both the exemplar fits and the gated spectra have their long-lived components removed before further analysis. The number system in this section makes it vague.

Section 3.5.1 – A figure in the supporting documents showing a raw spectrum and the long-lived component would be helpful in demonstrating how large this component is relative to the total signal. Or state it explicitly in the text.

Section 3.5.2 – How sensitive is the shape of the HCHO exemplar to the concentration of HCHO used? Why use the profile with 25 ppbv HCHO, instead of an average of several concentrations (scaled, presumably)?

P7L29 – What is the typical agreement between the two detection axes? State this.

Section 3.5.3 – Consider making a figure for the supporting documents showing a

spectrum with shading to indicate where the gating occurs.

P10L11 – It's not clear where the 100 pptv value comes from. . .

Section 5 – Is there any reason to believe the "air exemplars" or the long-lived component might be different in-flight than on the ground? How consistent are the auxiliary measurements (i.e. detector internal pressure, laser head temperature) during vertical profiles?

Figures 4 and 5 – What concentrations of HCHO do these examples correspond to? Can you put this information either on the plot or in the caption?

Table 1 – This table (and possibly Figure 11) could move to the supporting documents, as the focus of this paper is really on the instrument technique, and not on the measured spatial distribution of formaldehyde. You could give the details for the two flights in the caption for Figure 12 and leave out the rest.

Technical comments:

P3L8 – "In practice, the laser is turned on. . ." is odd phrasing, as it implies that something else was supposed to happen in theory.

P6L20 – "The fluorescence signal at the end of the bin-resolved data (∼400 ns after the laser pulse). . ." Change to >400 ns, since you are taking the signal from 400-500 ns.

P6L24 – State what times bins 75-100 correspond to. Same comment for bins 87-100 on P6L27.

---

## Referee Comment (RC2) · A. Fried (Referee) · 17 Sep 2017

This is an excellent paper, which describes a new extremely small and light weight instrument for airborne measurements of formaldehyde. The paper is well written and the procedures and methods employed are very sound and well documented. All aspects of the measurement technique, from detection principles, optics, sample handling, and especially data acquisition and processing, are described in very nice detail. The authors are to be commended for a very thorough instrument characterization. This reviewer recommends publication without any major changes. The authors may wish to consider the 3 very minor points discussed below:

[Figure]

1. It's clear from paper that the new COFFEE instrument is more susceptible to scattered light from aerosols than ISAF, and this is dramatically illustrated in Fig. 9. Although ISAF is not the focus of the present study, as the measurements from COFFEE are an outgrowth of ISAF measurements and analysis, the authors may wish to comment on what the ISAF signals look like with particle filtering, if available. Also, it would be worth commenting if Drierite-filtered air was added to the inlet in flight, or is this not necessary from your lab testing?

2. The authors on page 7 lines 28-29, indicate that "the final HCHO mixing ratio data product is produced by averaging the data from the two detection axes", it would be useful if the authors indicate what type of averaging was employed. Was a linear or weighted average employed? This could be important since as indicated on page 3 that detection axis 2 maximizes detection selectivity at the expense of deceased sensitivity, while axis 1 collects more fluorescence signal, potentially at the expense of selectivity. How are both attributes reflected in the final measurement?

3. My final minor comment regards the terminology of Fig. 7 and its discussion in the manuscript involving the Allan deviation plot. Although David Allen first introduced this concept in 1966 for characterization of frequency standards, it was Peter Werle's seminal paper (P. Werle, R. Mücke, and F. Slemr, "The Limits of Signal Averaging in Atmospheric Trace-Gas Monitoring by Tunable Diode-Laser Absorption Spectroscopy (TDLAS)", Applied Physics B57, 131-139, 1993), that first brought this valuable tool to the attention of the atmospheric measurement community. Several of us are trying to acknowledge Peter's legacy in the literature by now referring such analysis and plots as "Allan-Werle" plots and analysis.

Please also note the supplement to this comment:
https://www.atmos-meas-tech-discuss.net/amt-2017-282/amt-2017-282-RC2-supplement.pdf

---

## Author Comment (AC1) · 2 Oct 2017

The authors thank the two reviewers for their thoughtful comments. A reply to each comment, including changes to the manuscript, is included below with the original comment in *italic* font and the reply in normal font.

*Referee #1:*

*Section 3.3 – Can you address what, if any, losses of HCHO are expected on the inline particle filter (or even on the pressure controller)? Has this been tested? Were any laboratory calibrations conducted with the particle filters in place to make sure they match calibrations without filters?*

**Author reply:**
The transmission of HCHO by the particle filter was tested with the calibration system and found to be 100%. We have seen no evidence of HCHO transmission issues with the pressure controller. The COFFEE pressure controller includes features that should improve its time response to changing [HCHO] relative to ISAF (Carzorla et al): the actuator has less exposed surface area, and the valve block is nickel-plated.

We added the following line to Section 3.3 to address this question:

> The transmission of HCHO by the particle filter was tested with the calibration system and found to be 100%.

*P5L13 – Particle filters were used when "high aerosol loading is expected". Does this mean they were used in any non-laboratory deployment?*

**Author reply:**
In practice, the filter has been used for all AJAX flights. The qualifying text was added to convey that flights with low aerosol loading (e.g. no boundary layer sampling) could operate without the filter and avoid the small loss of conductance from the filter. To clarify, the text has been changed to:

> The instrument is operated with an inline particle filter (Balston 9922-05-DQ) to minimize related measurement artifacts from high aerosol loading (see Sect. 4.5).

*P5L15 – "The element retains 93% of the particles with a 0.01 um diameter." Does this mean it retains 93% of all particles larger than 0.01 um? Or is there some diameter over which it retains close to 100% of particles? What percentage of particle smaller than 0.01 um are retained, and do you expect this to make a difference in your analysis?*

**Author reply:**
The statement is how the manufacturer describes their filter performance, with a retention

number for a specific particle diameter. Unfortunately they do not provide more detailed filter performance information. We recently had access to a CPC courtesy of the Jimenez Group from U. Colorado Boulder and sampled NASA DC-8 cabin air with and without the filter in place. The filter removed 99.98% of the particles.

Our objective for including the particle filter is to keep the prompt signal from Mie scattering sufficiently low that it doesn't dominate the signal time profile, and the current filter performs well in this regard. Versions of the filter with higher particle retention are available if we discover that is necessary, but currently we do not have evidence that higher retention is needed.

*Section 3.3 – Is the inlet system here similar to the one shown in Fig. 5 of Carzorla et al? If not, describe the inlet used here. Is there anything in addition to the particle filter?*

**Author reply:**
The inlet system is not the same as in Cazorla—it is a less sophisticated set-up. The Section 3.3 text that describes the inlet system:

> The current Alpha Jet inlet is a rear-facing stainless steel tube 9.5 mm OD (6 mm ID) that extends 17 cm beyond the bottom of the pod. 9.5 mm OD (6.35 mm ID) THV fluoropolymer tubing connects the inlet to the instrument chassis. The instrument is operated with an inline particle filter (Balston 9922-05-DQ) to minimize related measurement artifacts from high aerosol loading (see Sect. 4.5).

*Section 3.5 –Clarify in the beginning of this section that both the exemplar fits and the gated spectra have their long-lived components removed before further analysis. The number system in this section makes it vague.*

**Author reply:**
Currently the long-lived component is removed for the exemplar fits but not for the gated data processing, though this would be possible because of the 5 ns resolution binned data. Figures S2 and S3 are helpful in putting the size of the long-lived component into context—it is small and likely will always have a negligible effect on the gated data. The exemplar fitting routine, however, benefits from fitting a signal that decays to zero at the end of the time profile.

To better distinguish the exemplar fit data processing approach and the gated count approach, the following was added to the end of Sect. 3.5:

> In addition to the fitting-based data processing, HCHO mixing ratios can also be obtained from gated count data, as discussed in Sect. 3.5.3.

*Section 3.5.1 – A figure in the supporting documents showing a raw spectrum and the long-lived component would be helpful in demonstrating how large this component is relative to the total signal. Or state it explicitly in the text.*

**Author reply:**
Figures are a good idea for showing the contribution of the long-lived component. We added Figs. S2 and S3 that show time profiles and long-lived components for detection axis 2 and 1, respectively, for the same 1 s data period as in Fig. 5.

*Section 3.5.2 – How sensitive is the shape of the HCHO exemplar to the concentration of HCHO used? Why use the profile with 25 ppbv HCHO, instead of an average of several concentrations (scaled, presumably)?*

**Author reply:**
The shape of the HCHO fluorescence time profile is independent of the concentration of HCHO. The HCHO exemplar is derived from data with reasonably high (~25 ppbv) HCHO to ensure that HCHO dominates the shape of the time profile so that, when the air-only profile is subtracted, it is only a minor subtraction rather than a difference of two similarly large numbers. The high HCHO data is used to obtain the shape of the HCHO exemplar, but the corresponding calibration factor is obtained from the full calibration run, not just the high HCHO data.

To improve clarity, the following was added to the end of the "Obtaining the exemplars" subsection:

> Once the air and HCHO exemplars are obtained, they are used to fit laboratory calibration data with multiple HCHO concentrations using the two-step fit described below. The calibration factor unique to this HCHO exemplar is obtained from the linear regression of the HCHO added by the calibration system and the HCHO exemplar scaling factor, which is output of the fit.

*P7L29 – What is the typical agreement between the two detection axes? State this.*

**Author reply:**
The two axes generally agree well. We are still evaluating the utility of the two axes. To address the reviewer's point, we have added a figure (S4) to the supporting documents showing 60 s data from the two axes and a linear fit. The following text was also added:

> Data from the two axes generally agree well—Fig. S4 shows the cross plot of 60 s data from the two axes, along with a linear fit (slope = 0.98).

*Section 3.5.3 – Consider making a figure for the supporting documents showing a*

*spectrum with shading to indicate where the gating occurs.*

**Author reply:**
Figure S5 was added to show a time profile with the gated region shaded.

*P10L11 – It's not clear where the 100 pptv value comes from: : :*

**Author reply:**
We changed the text to improve clarity.  It now reads:

> Currently no in-flight zeroing is performed for COFFEE.  The 100 pptv term in
> the uncertainty is intended to account for any changes in the background signal
> over the ~2 hour duration AJAX flights.  To date, we have not observed long time
> constant or high HCHO offset behavior with the COFFEE instrument that would
> be solved by in-flight zeroing.

*Section 5 – Is there any reason to believe the "air exemplars" or the long-lived
component might be different in-flight than on the ground? How consistent are the
auxiliary measurements (i.e. detector internal pressure, laser head temperature) during
vertical profiles?*

**Author reply:**
By design, the "air exemplars" and the long-lived component are allowed to change in
amplitude during flight in order to improve the fit.  The engineering data in flight so far
have given us no reason to suspect the instrument is behaving differently in flight than on
the ground or in the lab.  For example, both the power meter internal to the laser and our
external power meter after the detection cell indicate the laser output is very stable
through the flight, including through altitude changes.

*Figures 4 and 5 – What concentrations of HCHO do these examples correspond to?
Can you put this information either on the plot or in the caption?*

**Author reply:**
The mixing ratios are 29 ppbv and 8 ppbv, respectively.  This information has been added
to the figure captions.

*Table 1 – This table (and possibly Figure 11) could move to the supporting documents,
as the focus of this paper is really on the instrument technique, and not on the measured
spatial distribution of formaldehyde. You could give the details for the two flights
in the caption for Figure 12 and leave out the rest.*

**Author reply:**

Good suggestion-- Table 1 is now in the supporting document.

*Technical comments:*
*P3L8 – "In practice, the laser is turned on: : :" is odd phrasing, as it implies that something else was supposed to happen in theory.*

**Author reply:**
We agree. The phrasing is now improved:

> In flight operation, the laser turns on at the beginning of the flight and off at the end, with no other interaction.

*P6L20 – "The fluorescence signal at the end of the bin-resolved data (_400 ns after the laser pulse): : :" Change to >400 ns, since you are taking the signal from 400-500 ns.*

**Author reply:**
Done. The text is now:

> The fluorescence signal at the end of the bin-resolved data (>400 ns) is small but non-zero…

*P6L24 – State what times bins 75-100 correspond to. Same comment for bins 87-100 on P6L27.*

**Author reply:**
Done. Bins 75-100 correspond to 370-500 ns and bins 87-100 correspond to 430-500 ns. Note these times are relative to the start of the acquisition, and the laser pulse occurs at 50 ns.

**Referee #2 (Alan Fried):**

*1. It's clear from paper that the new COFFEE instrument is more susceptible to scattered light from aerosols than ISAF, and this is dramatically illustrated in Fig. 9. Although ISAF is not the focus of the present study, as the measurements from COFFEE are an outgrowth of ISAF measurements and analysis, the authors may wish to comment on what the ISAF signals look like with particle filtering, if available. Also, it would be worth commenting if Drierite-filtered air was added to the inlet in flight, or is this not necessary from your lab testing?*

**Author reply:**
COFFEE is indeed much more susceptible to scattered light from aerosols than ISAF due

to the much higher (~25x) laser power for COFFEE.  For ISAF (Cazorla, et al., 2015), the 'full gate' data is affected by high aerosol loading (e.g. biomass burning plumes), but the prompt signal from scattering by particles is very effectively excluded by the 'delayed gate'.  The following was added to Section 4.5 to clarify:

> The ISAF HCHO measurement is much less sensitive to the presence of particles due to its much lower (~25x) laser power, and utilizing a delayed gate for signal sampling is sufficient to exclude any artifact from scattering by particles.

 Drierite-filtered air has not yet been used to provide an in-flight 'zero' for COFFEE, and the benefits/downsides of performing in-flight zeroing for these low flush-volume HCHO instruments is an active topic of discussion both within and outside our research group. Currently we believe, for these low flush-volume instruments, the downsides outweigh the benefits.

For COFFEE on the Alpha Jet, in comparison to ISAF operation on the NASA DC-8, there is a potentially greater benefit to in-flight zeroing due to the lower sample flows: COFFEE draws 2.3 sLm while ISAF, as currently configured on the DC-8 for the ATom campaign, subsamples ~4 sLm from a bypass flow of 14-30 sLm.  That said, we have not observed long time constant or high HCHO offset behavior with the COFFEE instrument that would be solved by in-flight zeroing.  Even with the lower 2.3 sLm sample flow, the COFFEE instrument flush time is rapid (170 ms, as discussed in Section 4.4). Ultimately, the 100 pptv term in the stated COFFEE measurement uncertainty is intended to conservatively account for any changes in the background signal over the ~2 hour duration AJAX flights.

We changed the text in Section 4.3 to read:

> Currently no in-flight zeroing is performed for COFFEE.  The 100 pptv term in the uncertainty is intended to account for any changes in the background signal over the ~2 hour duration AJAX flights.  To date, we have not observed long time constant or high HCHO offset behavior with the COFFEE instrument that would be solved by in-flight zeroing.

*2. The authors on page 7 lines 28-29, indicate that "the final HCHO mixing ratio data product is produced by averaging the data from the two detection axes", it would be useful if the authors indicate what type of averaging was employed. Was a linear or weighted average employed? This could be important since as indicated on page 3 that detection axis 2 maximizes detection selectivity at the expense of deceased sensitivity, while axis 1 collects more fluorescence signal, potentially at the expense of selectivity. How are both attributes reflected in the final measurement?*

**Author reply:**
This is a very good point, and one that is not yet fully resolved.  Currently, for simplicity, we do an unweighted arithmetic mean of the two channels, but have considered other

options. One approach we have tried is averaging both channels to 1 minute data, taking the difference, and treating the difference as an 'error' term to be added to axis 1 before averaging the 1 s data from the two channels. Ultimately more field intercomparison data with ISAF or another HCHO instrument is necessary to decide which approach is superior. To improve clarity, the sentence now reads:

> The final HCHO mixing ratio data product is the arithmetic mean of the data from the two detection axes.

*3. My final minor comment regards the terminology of Fig. 7 and its discussion in the manuscript involving the Allan deviation plot. Although David Allen first introduced this concept in 1966 for characterization of frequency standards, it was Peter Werle's seminal paper (P. Werle, R. Mücke, and F. Slemr, "The Limits of Signal Averaging in Atmospheric Trace-Gas Monitoring by Tunable Diode-Laser Absorption Spectroscopy (TDLAS)", Applied Physics B57, 131-139, 1993), that first brought this valuable tool to the attention of the atmospheric measurement community. Several of us are trying to acknowledge Peter's legacy in the literature by now referring such analysis and plots as "Allan-Werle" plots and analysis.*

**Author reply:**
We are happy to recognize Peter Werle's contribution, and have changed the text to use "Allan-Werle" throughout.